# A Review of Racial Differences and Disparities in ECG

**DOI:** 10.3390/ijerph22030337

**Published:** 2025-02-25

**Authors:** Jianwei Zheng, Chizobam Ani, Islam Abudayyeh, Yunfan Zheng, Cyril Rakovski, Ehsan Yaghmaei, Omolola Ogunyemi

**Affiliations:** 1Department of Preventive and Social Medicine, Charles R. Drew University of Medicine and Science, Los Angeles, CA 90059, USA; 2Internal Medicine Department, Charles R Drew University of Medicine and Science, Los Angeles, CA 90059, USA; 3Internal Medicine Department, David Geffen School of Medicine, University of California Los Angeles, Los Angeles, CA 90024, USA; 4Veterans Administration Loma Linda Hospital, Loma Linda, CA 92357, USA; 5Department of Mathematics, University of California Los Angeles, Los Angeles, CA 90024, USA; 6Department of Mathematics, Schmid College of Science and Technology, Chapman University, Orange, CA 92886, USA

**Keywords:** racial disparities, ECG, healthcare inequities, social determinants of health, machine learning, cardiovascular disease, artificial intelligence

## Abstract

The electrocardiogram (ECG) is a widely used, non-invasive tool for diagnosing a range of cardiovascular conditions, including arrhythmia and heart disease-related structural changes. Despite its critical role in clinical care, racial and ethnic differences in ECG readings are often underexplored or inadequately addressed in research. Variations in key ECG parameters, such as PR interval, QRS duration, QT interval, and T-wave morphology, have been noted across different racial groups. However, the limited research in this area has hindered the development of diagnostic criteria that account for these differences, potentially contributing to healthcare disparities, as ECG interpretation algorithms largely developed from major population data may lead to misdiagnoses or inappropriate treatments for minority groups. This review aims to help cardiac researchers and cardiovascular specialists better understand, explore, and address the impact of racial and ethnic differences in ECG readings. By identifying potential causes—ranging from genetic factors to environmental influences—and exploring the resulting disparities in healthcare outcomes, we propose strategies such as the development of race-specific ECG norms, the application of artificial intelligence (AI) to improve diagnostic accuracy, and the diversification of ECG databases. Through these efforts, the medical community can advance toward more personalized and equitable cardiovascular care.

## 1. Introduction

Cardiovascular disease (CVD) remains one of the leading causes of morbidity and mortality worldwide. The prevalence of total CVD nearly doubled from 271 million in 1990 to 523 million in 2019, and the number of CVD deaths steadily increased from 12.1 million in 1990, reaching 18.6 million in 2019 [1]. Early and accurate diagnosis of CVD is crucial for effective treatment and management [2,3], and the electrocardiogram (ECG) plays a pivotal role in this process. The ECG is a non-invasive diagnostic tool used to assess heart function, detect arrhythmias, monitor heart conditions, and evaluate the electrical activity of the heart [4,5]. It is widely used in clinical settings due to its cost-effectiveness and ability to provide rapid, real-time information about heart health [6].

Despite the widespread use of ECGs, significant racial and ethnic disparities persist in the diagnosis, treatment, and outcomes of cardiovascular diseases. Certain racial groups experience disproportionately higher rates of CVD, including heart disease and stroke. Among CVD patients, Black individuals tend to have a higher burden of comorbidities and face an elevated risk of arrhythmia-related adverse outcomes compared to White individuals [7,8,9]. Similarly, Asian populations display different cardiovascular risk profiles, such as a higher prevalence of diabetes, hypertension, and narrower coronary arteries, factors which affect left ventricular function differently compared to Caucasians [10,11,12,13]. These racial and ethnic disparities underscore the need to better understand and address these differences, as they significantly affect clinical outcomes and contribute to the overall socioeconomic burden. These disparities are compounded by differences in access to healthcare, quality of treatment, and early diagnosis, leading to worse outcomes for marginalized racial groups. For example, healthcare inequalities linked to racial and ethnic disparities have been estimated to cost $1.24 trillion in the United States [14], including the cost of premature death and lower quality of care. The authors derived this figure through three analyses [15]: (1) direct medical costs using data from the Medical Expenditure Panel Survey (2002–2006) to compare healthcare spending across groups; and (2) indirect costs assessing productivity losses and potential savings from eliminating disparities with the same dataset; and (3) costs of premature death: Calculating the economic impact of early mortality using data from the National Vital Statistics Reports (2003–2006).

One critical area where these disparities manifest is in the interpretation of ECG data. Although the ECG is a standard diagnostic tool, several studies have indicated that certain ECG parameters, such as heart rate, QT interval, and repolarization patterns, may differ among racial and ethnic populations [16,17,18]. These differences raise concerns about potential healthcare disparities, particularly in how ECGs are interpreted and applied across diverse populations [19,20]. Moreover, these differences are often under-recognized, leading to misdiagnosis or delayed treatment, particularly in underrepresented groups. Additionally, the lack of diversity in ECG research and clinical studies has contributed to a lack of representation of certain racial groups in the development of ECG interpretation guidelines, further exacerbating the issue. Addressing these racial disparities in ECG data and cardiovascular care is essential to improving health outcomes and reducing health inequities [16,21,22]. There is a growing need to explore and understand the role of racial differences in ECG readings and to develop solutions that ensure more accurate, equitable healthcare for all patients. This review aims to examine the existing literature on racial differences in ECG data, highlight the healthcare disparities that arise from these differences, and explore potential solutions to mitigate these disparities through innovative approaches in ECG interpretation, machine learning, and more inclusive clinical practices.

## 2. Methods

A comprehensive literature search was conducted across multiple academic databases to capture a wide range of relevant studies. The databases searched included PubMed, Google Scholar, Scopus, and Web of Science.

The literature search was carried out using a combination of keywords and phrases related to racial differences in ECG data and cardiovascular health disparities. Key search terms included: “racial disparities ECG”; “electrocardiogram differences race”; “heart disease diagnosis racial differences”; “ECG interpretation racial groups”; “cardiovascular disparities electrocardiogram” and “social determinants of health ECG”.

Studies were included in the review if they met the following criteria: published in peer-reviewed journals between 1994 and 2024; focused on racial or ethnic differences in ECG data, cardiovascular disease outcomes, or healthcare disparities related to ECG interpretation; studies involving human populations or clinical data and written in English. Exclusion criteria included: studies not focused on racial or ethnic disparities in ECG data or cardiovascular health; non-peer-reviewed sources such as abstracts or conference posters; and studies with insufficient data or methodology that could not be reliably analyzed.

## 3. Racial Differences in ECG

Studies examining racial and ethnic differences in ECG reveal significant disparities that may affect clinical decision-making and outcomes. Table 1 provides a detailed overview of key studies exploring these differences, highlighting the heterogeneity in ECG findings across racial groups. Research indicates that African Americans may exhibit higher QRS voltages and longer QRS durations compared to Caucasians, potentially impacting the identification of left ventricular hypertrophy (LVH) in clinical settings. For example, Study 3 in Table 1 indicates persistent adverse trajectories in ECG LVH patterns, particularly among Black and Native American males. Differences in QT intervals across racial groups have been documented, with shorter QT intervals reported in African American and Asian populations compared to Caucasians. Study 8 identifies gender-specific variations within Asian populations when compared to Whites. Patterns such as benign early repolarization (ER) and T-wave inversions are more prevalent among African Americans. Studies 5 and 6 reveal that anterior T-wave inversion (TWI) and associated J-point elevations, common in healthy Black populations, may be misinterpreted as pathological findings.

While some studies have explored racial and ethnic differences in ECG readings, the research in this area is relatively sparse. These findings raise concerns about the application of generalized ECG standards in diverse populations. Disparities in ECG underscore the necessity of establishing population-specific norms to improve diagnostic accuracy and equity in cardiovascular care. Further large-scale, multi-ethnic studies are critical to enhance the representativeness and applicability of ECG criteria globally. Without a stronger foundation of research, the current reliance on generalized ECG standards can perpetuate disparities in diagnosis and treatment.

## 4. Potential Causes of Racial Differences in ECG

Several factors may contribute to the observed racial and ethnic differences in ECG readings. Genetic variations, particularly in ion channel function and cardiac conduction pathways, likely play a significant role. For instance, certain gene variants associated with ion channel regulation may be more prevalent in specific racial groups, impacting ECG waveforms [31,32]. Racial and ethnic groups often differ in lifestyle habits, such as diet, physical activity, and levels of chronic stress, all of which may influence ECG readings [33,34]. Socioeconomic disparities can also exacerbate these factors, with marginalized groups often having less access to healthcare and preventive services [35]. The higher prevalence of hypertension, diabetes, and obesity in certain racial groups, particularly African Americans and Hispanics [36,37], can also contribute to distinct ECG patterns [38]. These conditions influence heart function, and thus, ECG readings. Because many ECG algorithms and reference ranges are derived predominantly from Caucasian populations [16], a measurement bias may occur, resulting in skewed interpretations when applied to non-Caucasian groups.

## 5. Consideration of Gender and Genetic Contributions to ECG Variation

In addition to racial and ethnic factors, sex-based differences play a critical role in shaping ECG parameters and should be considered when evaluating cardiovascular diagnostics. Sex hormones—including estrogen, testosterone, and progesterone—are known to influence key ECG intervals such as the QT and PR intervals as well as T-wave morphology [39,40]. For example, estrogen has been associated with a prolonged QT interval, whereas testosterone may exert a shortening effect [41,42]. These hormonal influences contribute to distinct electrophysiological profiles between males and females, which may interact with racial factors to further modulate ECG readings.

Genetic differences linked to sex also contribute to these variations. Sex chromosome-related factors, such as X-inactivation in females and Y-chromosome effects in males, can impact the expression of cardiac ion channels—including SCN5A and KCNH2—thereby altering cardiac conduction and repolarization [43,44]. Such genetic influences may underlie some of the observed differences in ECG parameters across diverse populations.

Furthermore, hormonal changes associated with life stages—for instance, the transition into menopause—may exacerbate these differences and have significant clinical implications for cardiovascular risk assessment and diagnosis [45]. Integrating these sex-based biological factors with racial and ethnic considerations is essential for refining diagnostic criteria and advancing personalized cardiovascular care.

## 6. Healthcare Disparities Stemming from Racial Differences in ECG

The variations in ECG parameters among racial groups have significant clinical consequences that can directly influence diagnosis and treatment decisions, further contributing to healthcare disparities [16]. Relying on standard ECG criteria—largely derived from predominantly Caucasian populations—can lead to misinterpretations that adversely affect patient management. For example, the higher prevalence of early repolarization in African Americans is often misinterpreted as myocardial infarction, leading to unnecessary interventions, such as inappropriate cardiac procedures [26]. Conversely, conditions like long QT syndrome may be underdiagnosed in African Americans due to the generally shorter QT intervals observed in this population. Reliance on standardized ECG reference values, which may not account for these racial differences, can result in inappropriate medical decisions, such as mismanagement of antiarrhythmic therapies, potentially harming patients by either overcorrecting or neglecting their actual conditions [16]. Studies have shown that conventional ECG criteria for diagnosing left ventricular hypertrophy (LVH) may have reduced sensitivity in African American patients, leading to underdiagnosis and delayed therapeutic intervention [46]. Similarly, differences in repolarization patterns among racial groups have been associated with variability in diagnosing myocardial ischemia, potentially resulting in either overestimation or underestimation of cardiovascular risk [14].

Real-world examples further underscore these concerns. In one reported case, an African American patient with atypical ECG findings was initially misdiagnosed with benign early repolarization, which delayed the identification and treatment of an underlying cardiomyopathy [47]. Other case studies have documented instances in which misinterpretation of ECG abnormalities in Hispanic patients led to unnecessary invasive procedures, as well as missed opportunities for timely intervention [48].

## 7. Innovative Solutions to Address Disparities and Future Directions

One solution lies in developing race-specific ECG reference values [49], particularly for key parameters like the QT interval and QRS duration. This could enable more accurate diagnoses and reduce the risk of misinterpretation [50]. Moreover, Machine Learning (ML) models offer a promising avenue for improving ECG interpretation by accounting for racial and ethnic differences [23,24,51]. By training models on racially diverse datasets, AI could provide tailored interpretations that are more reflective of the patient’s background. A critical step is to diversify ECG datasets used for developing diagnostic algorithms. Many ECG databases, such as the PTB XL [52], MIMIC Waveform [53], and Chapman ECG [54], are predominantly based on Caucasian populations or specific racial groups. Including more racial and ethnic groups will improve the generalizability of these tools. Healthcare providers should be trained to recognize racial differences in ECG patterns. Continued education programs should incorporate these nuances to improve clinicians’ ability to provide accurate and personalized care. Updating clinical guidelines to include race-specific ECG norms will help standardize care and ensure that racial differences are considered in routine clinical practice.

While these solutions offer a path forward, several challenges remain. First, the underrepresentation of racial and ethnic minorities in clinical research continues to be a major barrier [55]. Without adequate representation, it will be difficult to develop accurate, race-specific reference ranges. Additionally, while race can be a useful proxy for genetic, environmental, and social factors, it is not a perfect marker. Future studies should aim to disentangle the biological, environmental, and social determinants of racial differences in ECG data [56,57], focusing on individual characteristics rather than relying solely on racial categorization. While ML offers promising avenues for improving the accuracy of ECG interpretations and reducing racial disparities in cardiovascular diagnostics, it is important to acknowledge its limitations—particularly the risk of perpetuating existing biases. Many ML models are developed using training datasets that are not adequately representative of diverse racial groups. When such imbalances exist, the models may inadvertently learn and reinforce historical biases, leading to differential performance across populations [58,59]. This limitation is critical because an ML model trained predominantly on data from one racial group may not generalize well to others, potentially resulting in misdiagnosis or inappropriate treatment recommendations for underrepresented populations.

## 8. Conclusions

This review highlights the significant impact of racial and ethnic disparities in ECG interpretation and the consequent implications for cardiovascular care. While current strategies such as AI-driven diagnostic tools and race-specific ECG norms show promise, our analysis underscores the following future research directions to address these disparities.

Longitudinal Studies:We advocate for longitudinal studies that track ECG parameters across diverse racial groups over extended periods. Such studies will be critical to understanding how disparities evolve over time and in response to interventions, providing insights into the long-term clinical outcomes associated with these differences.

2.Diverse and Representative Datasets:There is a pressing need to develop large, racially balanced ECG databases. Future research should focus on curating and sharing datasets that accurately reflect the demographic diversity of the population. These datasets will be essential for training ML models that are robust and generalizable across different racial groups.

3.Integrated Analysis of Sex and Race:Future investigations should integrate analyses of sex-based differences with racial disparities in ECG readings. Research examining the interplay between hormonal, genetic, and environmental factors will help refine diagnostic criteria and ensure more personalized treatment approaches.

4.Validation of Race-Specific Norms:Prospective, multicenter trials are needed to validate race-specific ECG norms. Such studies should assess the clinical utility of these norms in reducing misdiagnosis and improving treatment decisions, thereby translating research findings into practice.

5.Bias Mitigation in ML:Future work should also focus on developing and implementing bias detection and correction methods in ML models used for ECG interpretation. Continuous evaluation and adaptation of these models in real-world settings are imperative to ensure that advancements in AI lead to equitable healthcare outcomes.

By pursuing these research avenues, we can better understand the underlying causes of ECG disparities and develop more effective, equitable diagnostic tools. These steps will not only enhance our scientific knowledge but also drive meaningful improvements in clinical practice, ultimately reducing the burden of cardiovascular disease across all racial and ethnic groups.

## Figures and Tables

**Table 1 ijerph-22-00337-t001:** Key Studies on Racial Differences in ECG.

#	Racial/Ethnic Groups Studied	Study Design and Populations	ECG or ECG Measurement Investigated	Associated Cardiac Conditions	Key Findings	Reference and Published Year
1	Non-Hispanic White; Asian; Black/African American; Hispanic/Latino; American Indian/Native Alaskan	A retrospective cohort analysis that included 97,829 patients with paired ECGs and echocardiograms collected by Mayo Clinic	12 lead ECG	Low left ventricular ejection fraction	ECG racial variations did not impact the ability of a convolutional neural network to predict low left ventricular ejection fraction from the ECG	Noseworthy, P. A., et al. (2020) [23]
2	Non-Hispanic White; Hispanic; Black and Asian	A retrospective analysis used 12-lead ECGs taken between 2008 and 2018 from 326,518 patient encounters referred for standard clinical indications to Stanford Hospital	12-lead ECG	Heart Failure	There were no significant differences observed between racial groups overall. However, the primary model performed significantly worse in Black patients aged 0 to 40 years compared with all other racial groups in this age group, with differences most pronounced among young Black women.	Kaur, Dhamanpreet, et al. (2024) [24]
3	White; Black; Asian and Pacific Islander; Latino; Mixed; and Native American	Kaiser Permanente Northern California’s network collected from the Northern California population.	Left ventricular hypertrophy (LVH) was measured by Cornell voltage-duration product.	Left ventricular hypertrophy	Adverse trajectories of ECG LVH (persistent, new development, or variable pattern) were more common among Blacks and Native American men and were independently related to incident cardiovascular disease with hazard ratios ranging from 1.2 for ECG LVH variable pattern and transient ischemic attack in women to 2.8 for persistent ECG LVH and heart failure in men.	Iribarren, Carlos, et al. (2017) [21]
4	Chinese; Nigerian; Black and Caucasians	The ECG data were available for four population samples gathered in Scotland, Taiwan, Nigeria and India.	QRS voltages and ST amplitudes	Non specified	QRS voltages were higher in young Black males compared to others, while ST amplitudes, as in V2, were higher in Chinese and Nigerian males than in Caucasians	Macfarlane, P. W., et al. (2014) [25]
5	Black adults	Review study for healthy Black adults	QRS voltage; Early repolarization; T wave inversion; anterior STE and TWI with associated J point elevation	Healthy Black adults	Six ECG patterns are found more frequently in healthy Black adults than in Whites.	Walsh, B., et al. (2019) [26]
6	North American White; Black and Hispanic	A retrospective analysis used 12-lead ECGs from Second National Health and Nutrition Examination Survey and the Hispanic Health and Nutrition Examination Survey	ECG amplitudes	Ethnic differences in ECG amplitudes	There were substantial racial differences in ECG amplitudes. In general, ECG amplitudes and amplitude combinations used in left ventricular hypertrophy (LVH) criteria were larger in Blacks than inWhites.	Rautaharju, P. M., et al. (1994) [16]
7	White; African American; Hispanic and Chinese	A cross-sectional analysis in the MESA (Multi-Ethnic Study of Atherosclerosis), a community-based cohort study that enrolled 6814 Americans free of clinically recognized cardiovascular disease in 2000 to 2002.	AF	AF	The prevalence of clinically detected AF after 14.4 years’ follow-up was 11.3% in whites, 6.6% in African Americans, 7.8% in Hispanics, and 9.9% in Chinese and was significantly lower in African Americans than in Whites, in both unadjusted and risk factor-adjusted analyses. By contrast, in the same individuals, the proportion with monitor-detected AF using a 14-day ambulatory ECG monitor was similar in the 4 race/ethnic groups: 7.1%, 6.4%, 6.9%, and 5.2%, respectively (compared with Whites, all *p* > 0.5).	Heckbert, S. R. et al. (2020) [27]
8	White and Asian	We studied 2677 White Framingham Heart Study participants and 2972 Asian from Singapore Longitudinal Aging Study participants free of myocardial infarction or heart failure.	P-wave; LVH; LAE; QTc; PR interval, QRS duration, QT interval, QRS voltage	Free of myocardial infarction or heart failure.	PR interval was longer in Asians compared with Whites. QT interval was shorter in Asian men and longer in Asian women compared to White men and women, respectively. Asians had greater odds of having ECG left ventricular hypertrophy (LVH) compared with Whites.	Santhanakrishnan, R., et al. (2016) [22]
9		The distribution of baseline measures of were compared by ethnicity in 15,429 participants (27% Black) from the Atherosclerosis Risk in Communities (ARIC) study.	P-wave terminal force, P-wave duration, P-wave area, and PR duration	AF	AF was significantly less common in Blacks compared with Whites. Black ethnicity was significantly associated with abnormal AF predictors compared with whites. AF predictors were significantly and independently associated with AF and ischemic stroke with no significant interaction between ethnicity and AF predictors, findings that further justify using AF predictors as an earlier indicator of future risk of AF and stroke.	Soliman, E. Z et al. (2009) [28]
10	Black and White	2463 Black and White patients with heart failure and left ventricular ejection fraction ≤ 35% who underwent coronary angiography and 12-lead electrocardiography at Duke University Hospital from 1995 through 2011.	Prolonged QRS duration	Left Ventricular Systolic Dysfunction	QRS duration was longest in White men followed by White women, Black men and Black women. Left bundle branch block was more common in women than men and in White versus Black individuals.	Tiffany C. Randolph et al. (2017) [29]
11	African American and White	Participants were 148 employed men and women between the ages of 25 and 45 years who participated in the Duke Biobehavioral Investigation of Hypertension (BIOH).	Heart rate variability	Increased left ventricular mass and Hypertension	Greater high-frequency heart rate variability (HF-HRV) was associated with greater Increased left ventricular mass (LVM) among African Americans but was not related to LVM in Whites	Hill, LaBarron K., et al. (2017) [30]

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
