# Peer review of "A Review of Racial Differences and Disparities in ECG"

_ijerph, 2025, doi:10.3390/ijerph22030337_

Round 1
Reviewer 1 Report
Comments and Suggestions for Authors
With their work, the authors present a review that analyzes publications regarding racial differences in ECG-related diagnostic parameters. They focused on parameters such as PR interval, QRS duration, QT interval and T-wave morphology. For this purpose, based on a search in PubMed, Google Scholar, Scopus, and Web of Science, a total of 11 selected studies were used, which were published between 1994 and 2024. The authors conclude that racial disparities in this regard are well known and see the need to address health care inequalities in diagnosis and treatment. They see potential for improvements here through personalized medicine and AI-controlled diagnostic options as well as through diversification of training data sets and development of breed-specific references.
From the reviewer's perspective, it should be explained in more detail how one arrives at the cost of "$1.24 trillion" in lines 52-53 ("healthcare inequalities linked to racial and ethnic disparities have been estimated to cost $1.24 trillion in the United States, including the cost of premature death and lower quality of care").
Author Response
Comments 1: From the reviewer's perspective, it should be explained in more detail how one arrives at the cost of "$1.24 trillion" in lines 52-53 ("healthcare inequalities linked to racial and ethnic disparities have been estimated to cost $1.24 trillion in the United States, including the cost of premature death and lower quality of care").
Response 1: Thank you for your insightful comment regarding the derivation of the $1.24 trillion estimate. In response, we have expanded the explanation in the revised manuscript to provide additional detail on this calculation. Specifically, the estimate is derived from an analysis that aggregates three components highlighted in the revised manuscript (Line53-59). Each component is explained with reference to specific datasets (MEPS for direct and indirect costs and National Vital Statistics Reports for premature death), providing a comprehensive view of the methodology used. We believe these revisions enhance the clarity and rigor of our economic analysis and thank the reviewer for helping us improve the manuscript.

Reviewer 2 Report
Comments and Suggestions for Authors
The manuscript by Zheng et al. provides a comprehensive review of racial and ethnic disparities in electrocardiogram (ECG) readings and their implications for cardiovascular care. The authors discuss variations in ECG parameters such as PR interval, QRS duration, QT interval, and T-wave morphology across different racial groups, highlighting the potential for misdiagnosis due to reliance on ECG interpretation algorithms developed from predominantly Caucasian populations. The paper explores genetic, environmental, and socioeconomic factors contributing to these disparities and proposes solutions such as the establishment of race-specific ECG norms, the use of artificial intelligence (AI) to improve diagnostic accuracy, and diversification of ECG datasets to better represent minority populations. The authors emphasize the need for personalized and equitable cardiovascular diagnostics and care.
While this review provides valuable insights into an underexplored aspect of cardiovascular research, there are some major and minor issues that should be addressed to strengthen the manuscript.
Major point
1. Insufficient Consideration of Sex-Based Differences in ECG Racial Disparities
The manuscript does not adequately address sex-based differences within racial groups in ECG parameters. Sex hormones (estrogen, testosterone, progesterone) influence QT interval, PR interval, and T-wave morphology, with effects varying across racial groups. Additionally, sex chromosome-linked differences (e.g., X-inactivation, Y-chromosome effects) impact ion channel expression and cardiac conduction. The authors should discuss how these biological factors contribute to racial ECG disparities, particularly in relation to hormonal changes (e.g., menopause) and genetic influences on ion channels (e.g., SCN5A, KCNH2), to provide a more comprehensive analysis of ECG differences across populations.
2. Lack of Discussion on the Clinical Impact of ECG Disparities on Treatment Decisions
Lack of Discussion on the Clinical Impact of ECG Disparities on Treatment Decisions The manuscript mentions racial differences in ECG but lacks real-world clinical examples of their impact on diagnosis and treatment. Including case studies of misdiagnoses or treatment errors due to ECG disparities would add valuable context. Additionally, discussing how these disparities influence treatment adjustments, such as incorrect therapies or missed diagnoses, would enhance clinical relevance.
3. Limited Discussion on Genetic Contributions to ECG Variability
The authors mention genetic factors but do not explore them in sufficient detail. Recent studies have identified specific genetic variants influencing cardiac conduction pathways and ECG patterns, particularly in ion channels and conduction system genes (e.g., SCN5A, KCNH2). Expanding this section with concrete genetic evidence, including a discussion of genome-wide association studies (GWAS), would enhance the manuscript’s scientific rigor.
Minor point
1. Citation of Outdated References
Several references cited are over 15–20 years old, which may not reflect the latest advancements in ECG disparities research. Replace with more recent studies that explore racial differences using contemporary ECG databases.
2. Limited Discussion on Machine Learning Limitations
The manuscript proposes machine learning (ML) models as a solution to ECG disparities but does not address bias in training datasets. Since many ML models are trained on datasets with racial imbalances, they may reinforce existing disparities instead of solving them. This limitation should be acknowledged.
3. The Conclusion Needs Stronger Future Directions
The conclusion briefly mentions AI and race-specific norms but does not provide concrete next steps for research. Adding a more detailed call to action for future research directions (e.g., longitudinal studies on racial ECG disparities) would make the conclusion more impactful.
4. Minor Grammatical and Formatting Issues
There are grammatical errors and awkward phrasings throughout the manuscript. For example, in the introduction:
"Studies have shown that certain racial groups are disproportionately affected by CVD, experiencing higher rates of heart disease, stroke, and related conditions." Line 40-42
This sentence is somewhat redundant. It can be revised as:
"Certain racial groups experience disproportionately higher rates of CVD, including heart disease and stroke."
Author Response
Comments 1: Insufficient Consideration of Sex-Based Differences in ECG Racial Disparities
The manuscript does not adequately address sex-based differences within racial groups in ECG parameters. Sex hormones (estrogen, testosterone, progesterone) influence QT interval, PR interval, and T-wave morphology, with effects varying across racial groups. Additionally, sex chromosome-linked differences (e.g., X-inactivation, Y-chromosome effects) impact ion channel expression and cardiac conduction. The authors should discuss how these biological factors contribute to racial ECG disparities, particularly in relation to hormonal changes (e.g., menopause) and genetic influences on ion channels (e.g., SCN5A, KCNH2), to provide a more comprehensive analysis of ECG differences across populations.
Response 1: Thank you for highlighting the need to incorporate sex-based differences in our analysis of ECG racial disparities. We agree that understanding how sex influences ECG parameters is crucial for a comprehensive review. To address this major issue, we have dedicated section on sex-Based differences (Page 6, Line 153-171).
Comments 2: Lack of Discussion on the Clinical Impact of ECG Disparities on Treatment Decisions
Lack of Discussion on the Clinical Impact of ECG Disparities on Treatment Decisions The manuscript mentions racial differences in ECG but lacks real-world clinical examples of their impact on diagnosis and treatment. Including case studies of misdiagnoses or treatment errors due to ECG disparities would add valuable context. Additionally, discussing how these disparities influence treatment adjustments, such as incorrect therapies or missed diagnoses, would enhance clinical relevance.
Response 2: Thank you for the constructive suggestions. We have re-written the section 6 (line 172-228) to address the clinical impact of ECG disparities on treatment decisions.
Comments 3: Limited Discussion on Genetic Contributions to ECG Variability
The authors mention genetic factors but do not explore them in sufficient detail. Recent studies have identified specific genetic variants influencing cardiac conduction pathways and ECG patterns, particularly in ion channels and conduction system genes (e.g., SCN5A, KCNH2). Expanding this section with concrete genetic evidence, including a discussion of genome-wide association studies (GWAS), would enhance the manuscript’s scientific rigor.
Response 3: Thank you for your insightful suggestion. We have added a new section "5. Consideration of Gender and Genetic Contributions to ECG Variation" (Page 6, Line 153-171) to address genetic variants linked to cardiac conduction pathways and ECG patterns.
Comments 4: Citation of Outdated References
Several references cited are over 15–20 years old, which may not reflect the latest advancements in ECG disparities research. Replace with more recent studies that explore racial differences using contemporary ECG databases.
Response 4: Thank you for your valuable feedback regarding the use of older references. We acknowledge that several citations in our manuscript are over 15–20 years old. In our comprehensive literature search, we discovered that recent studies specifically focusing on the racial disparities in ECG remain scarce. Although there have been significant advancements in ECG technology and analytical methodologies over the past decade, research directly addressing racial differences using contemporary ECG databases is still limited. To address this concern, we have taken the following steps:
-
Revised Literature Search:
We conducted an updated literature search to identify any recent studies that explore racial disparities in ECG. While a few newer studies touch on related aspects—such as advancements in machine learning-based ECG interpretation and the development of more diverse ECG datasets—the number of studies explicitly focused on racial differences remains limited. -
Manuscript Revision:
We have revised the manuscript to highlight this gap in the literature, emphasizing the need for more contemporary research in this area. This discussion now underscores the importance of our review and calls attention to the critical need for future studies using modern ECG databases. -
Supplementary Recent References:
Where possible, we have supplemented our reference list with more recent citations related to advancements in ECG technology and machine learning applications in cardiovascular diagnostics. These additions help contextualize our findings within the framework of current technological developments, even though direct studies on racial disparities remain sparse.
Comments 4: Limited Discussion on Machine Learning Limitations
The manuscript proposes machine learning (ML) models as a solution to ECG disparities but does not address bias in training datasets. Since many ML models are trained on datasets with racial imbalances, they may reinforce existing disparities instead of solving them. This limitation should be acknowledged.
Response 4: Thank you for raising this important concern regarding bias in machine learning (ML) models. We agree that ML models trained on racially imbalanced datasets may inadvertently reinforce existing disparities rather than resolving them. In response to your comment, we have revised the manuscript to explicitly acknowledge this limitation (Line 252 -261).
Comment 5: The Conclusion Needs Stronger Future Directions
The conclusion briefly mentions AI and race-specific norms but does not provide concrete next steps for research. Adding a more detailed call to action for future research directions (e.g., longitudinal studies on racial ECG disparities) would make the conclusion more impactful.
Response 5: Thank you for your valuable feedback regarding the conclusion. We agree that providing more concrete future directions will enhance the impact of our conclusions. In response, we have revised the conclusion to include a detailed call to action for future research. Specifically, we now emphasize the need for:
Longitudinal Studies: To track racial differences in ECG parameters over time and assess the long-term clinical outcomes of these disparities.
Diverse and Representative Datasets: To build and utilize comprehensive, racially balanced ECG databases for training and validating machine learning models.
Integrated Analyses: Combining sex-based and racial differences in ECG readings to better understand their interplay and refine diagnostic criteria.
Validation of Race-Specific Norms: Through prospective, multi-center trials to determine the clinical utility of tailored ECG reference values.
Bias Mitigation in ML: Focusing on developing and applying strategies to detect and correct bias in machine learning algorithms used for ECG interpretation.
Comments 6: Minor Grammatical and Formatting Issues
There are grammatical errors and awkward phrasings throughout the manuscript. For example, in the introduction:
"Studies have shown that certain racial groups are disproportionately affected by CVD, experiencing higher rates of heart disease, stroke, and related conditions." Line 40-42
This sentence is somewhat redundant. It can be revised as:
"Certain racial groups experience disproportionately higher rates of CVD, including heart disease and stroke."
Response 6: Thank you for highlighting the grammatical and formatting issues in our manuscript. We have carefully reviewed the text and made the necessary revisions to improve clarity and conciseness. We have applied similar revisions throughout the manuscript to eliminate redundancy and enhance overall readability. Thank you again for your constructive feedback!

Round 2
Reviewer 2 Report
Comments and Suggestions for Authors
Thank you for the revision. I have no further comments.